# Supervised machine learning to predict smoking lapses from Ecological Momentary Assessments and sensor data: Implications for just-in-time adaptive intervention development

Olga Perski [1,2,3]*, Dimitra Kale[3], Corinna Leppin[3], Tosan Okpako [3], David Simons [4], Stephanie P. Goldstein[5], Eric Hekler[2], Jamie Brown [3]

1 Faculty of Social Sciences, Tampere University, Finland, 2 Herbert Wertheim School of Public Health and Human Longevity Science, University of California, San Diego, California, United States of America, 3 Department of Behavioural Science and Health, University College London, United Kingdom, 4 Centre for Emerging, Endemic and Exotic Diseases, Royal Veterinary College, United Kingdom, 5 Department of Psychiatry and Human Behavior, Warren Alpert Medical School of Brown University & The Miriam Hospital/ Weight Control and Diabetes Research Center, United States of America

* olga.perski@tuni.fi

**Data Availability Statement:** The data and R code underpinning the analyses are openly available

## Abstract

Specific moments of lapse among smokers attempting to quit often lead to full relapse, which highlights a need for interventions that target lapses before they might occur, such as just-in-time adaptive interventions (JITAIs). To inform the decision points and tailoring variables of a lapse prevention JITAI, we trained and tested supervised machine learning algorithms that use Ecological Momentary Assessments (EMAs) and wearable sensor data of potential lapse triggers and lapse incidence. We aimed to identify a best-performing and feasible algorithm to take forwards in a JITAI. For 10 days, adult smokers attempting to quit were asked to complete 16 hourly EMAs/day assessing cravings, mood, activity, social context, physical context, and lapse incidence, and to wear a Fitbit Charge 4 during waking hours to passively collect data on steps and heart rate. A series of group-level supervised machine learning algorithms (e.g., Random Forest, XGBoost) were trained and tested, without and with the sensor data. Their ability to predict lapses for out-of-sample (i) observations and (ii) individuals were evaluated. Next, a series of individual-level and hybrid (i.e., group- and individual-level) algorithms were trained and tested. Participants (N = 38) responded to 6,124 EMAs (with 6.9% of responses reporting a lapse). Without sensor data, the best-performing group-level algorithm had an area under the receiver operating characteristic curve (AUC) of 0.899 (95% CI = 0.871–0.928). Its ability to classify lapses for out-of-sample individuals ranged from poor to excellent (AUC_per person = 0.524–0.994; median AUC = 0.639). 15/38 participants had adequate data for individual-level algorithms to be constructed, with a median AUC of 0.855 (range: 0.451–1.000). Hybrid algorithms could be constructed for 25/38 participants, with a median AUC of 0.692 (range: 0.523 to 0.998). With sensor data, the best-performing group-level algorithm had an AUC of 0.952 (95% CI = 0.933–0.970). Its ability to classify lapses for out-of-sample individuals ranged from poor to excellent (AUC_per

here: https://zenodo.org/doi/10.5281/zenodo.
10472386.

**Funding:** OP is supported by a Marie Skłodowska-
Curie Postdoctoral Fellowship from the European
Union (Grant Agreement number: 101065293).
Views and opinions expressed are however those
of the author(s) only and do not necessarily reflect
those of the European Union. The European Union
cannot be held responsible for them. DK, CL, and
JB receive salary support from Cancer Research
UK (PRCRPG-Nov21\100002). The funders had no
role in study design, data collection and analysis,
decision to publish, or preparation of the
manuscript.

**Competing interests:** OP and JB act as unpaid
scientific advisors to the Smoke Free app. JB has
received unrestricted research funding from Pfizer
to study smoking cessation.

$_{person}$ = 0.494–0.979; median AUC = 0.745). 11/30 participants had adequate data for indi-
vidual-level algorithms to be constructed, with a median AUC of 0.983 (range: 0.549–
1.000). Hybrid algorithms could be constructed for 20/30 participants, with a median AUC of
0.772 (range: 0.444 to 0.968). In conclusion, high-performing group-level lapse prediction
algorithms without and with sensor data had variable performance when applied to out-of-
sample individuals. Individual-level and hybrid algorithms could be constructed for a limited
number of individuals but had improved performance, particularly when incorporating sen-
sor data for participants with sufficient wear time. Feasibility constraints and the need to bal-
ance multiple success criteria in the JITAI development and implementation process are
discussed.

## Author summary

Among cigarette smokers attempting to stop, lapses (i.e., temporary slips after the quit
date) are common and often lead to full relapse (i.e., smoking as regular). The timing of
and reasons for lapses (e.g., stress, low motivation) differ from person to person. Despite
lapses being a key reason for full relapse, there is little dedicated support available to help
prevent them. This study used self-reported data from brief, hourly surveys sent directly
to people's smartphones in addition to passively collected data from smartwatches to train
and test group-level, individual-level, and hybrid (i.e., a combination of group- and indi-
vidual-level) lapse prediction algorithms. This was with a view to informing the develop-
ment of a future 'just-in-time adaptive intervention' (JITAI) that can provide personalised
support to smokers in real-time, when they most need it. We found that individual-level
and hybrid algorithms performed better than the group-level algorithms, particularly
when including the passively collected sensor data. However, multiple success criteria
(e.g., acceptability, scalability, technical feasibility) need to be carefully balanced against
algorithm performance in the JITAI development and implementation process.

## Introduction

Cigarette smoking is responsible for ~8 million global deaths each year [1]. Supporting smok-
ers to quit is a public health priority. About 40% of smokers make a quit attempt each year [2],
of whom <5% who quit unaided remain abstinent for one year [3]. Pharmacological or beha-
vioural support, delivered in-person or via digital interventions, can substantially improve the
odds of quitting [4–6]; however, absolute quit rates remain low due to temporary slips or
'lapses', which often set people on a course to regular smoking [7,8]. Brief, skills-based inter-
ventions do not help to prevent transitions from temporary lapses to full relapse [9], which
may at least partly be explained by corroborating evidence indicating that lapse risk fluctuates
over time (i.e., it is 'dynamic') and is influenced by person-specific internal and external factors
(i.e., it is 'idiosyncratic') [10–14]. More specifically, low frequency skills-based interventions
delivered at times of low risk may not be used later during moments of high lapse risk. This
has led to the development of real-time relapse prevention interventions, which aim to deliver
the right type of support to individuals at times when they most need it (i.e., 'just-in-time adap-
tive interventions'; JITAIs). According to Nahum-Shani and colleagues, JITAIs include the fol-
lowing building blocks: i) decision points (i.e., time points at which the delivery of an

**Table 1. Overview of prior smoking cessation JITAIs.**

| Authors | JITAI name | Theory- vs. data-driven approach | Detection vs. prediction | EMA vs. sensor data | Group-level vs. individual-level vs. hybrid algorithm |
|---|---|---|---|---|---|
| Naughton et al. 2016; Naughton et al. 2023 | Quit Sense | Theory-driven | Detection of user-specified high-risk location(s) | Sensor data (geofencing) | Individual-level |
| Businelle et al. 2016; Hébert et al. 2020 | Smart-T2 | Data-driven | Prediction of lapses | EMA | Group-level |
| Saleheen et al. (2015); Hovsepian et al. (2015); Battalio et al. (2021) | Sense2Stop | Data-driven | Detection of stress and lapses | Sensor data (suite of chest- and wrist-worn sensors) and EMA (not used to trigger support in current version) | Hybrid algorithm with parameters tuned to individual data over time |
| Saleheen et al. (2015); Hovsepian et al. (2015); Yang et al. (2023) | Time2Quit | Data-driven | Detection of stress and lapses | Sensor data (suite of chest- and wrist-worn sensors) and EMA (not used to trigger support in current version) | Hybrid algorithm with parameters tuned to individual data over time |

intervention might be beneficial), ii) tailoring variables (i.e., variables which provide information about when and how to intervene), iii) intervention options (i.e., change strategies or delivery options for consideration), and iv) decision rules (i.e., algorithms that link decision points, tailoring variables and intervention options) [15]. Although several JITAIs for smoking cessation have been developed (see Table 1 for an overview), a recent systematic review concluded that few available JITAIs have used a data-driven approach to inform the selection of decision points and tailoring variables [16]. In addition, few available JITAIs have harnessed wearable sensor data (e.g., step count, heart rate) to detect states of vulnerability. Therefore, this study aimed to develop and evaluate a series of group-level, individual-level, and hybrid supervised machine learning algorithms to predict momentary lapse risk in smokers attempting to stop, harnessing Ecological Momentary Assessments (EMAs; i.e., brief surveys in people's daily lives) and wearable sensor data. This was with a view to leveraging the best-performing and feasible algorithm within a subsequent smoking lapse prevention JITAI.

## Prior smoking cessation JITAIs

A key question for JITAI developers is knowing when to intervene for each individual (i.e., identifying decision points and tailoring variables) and what data source(s) to use for reliable inputs. Several approaches–primarily focused on theory-informed selection or predictive modelling–have been developed and tested (summarised in Table 1). Other approaches, which also tend to incorporate the identification of what intervention to provide at moments of need, place a greater emphasis on causal modelling. These approaches include experimental designs (e.g., micro-randomised trials, system identification experiments from control engineering) and formal, dynamical systems modelling [17–20]. We do not expand further on such approaches in the present study as it was not our intention to identify what intervention to provide at times of predicted need (although see [21,22] for micro-randomised trials in progress within the smoking cessation domain). For example, Naughton and colleagues developed a theory-informed smoking cessation JITAI which used geofencing technology to *detect* (rather than *predict*) entry into user-specified, high-risk smoking locations [23,24]. Using a data-driven approach, Businelle and colleagues collected frequent EMAs and trained and tested a group-level supervised machine learning algorithm, which was embedded within a subsequent JITAI [11,25]. However, the ability of the group-level algorithm to accurately *predict* lapse risk for each individual remains an empirical question.

Recently, a hybrid (i.e., group- and individual-level) algorithm was developed to *predict* lapses from a recommended diet in behavioural weight loss treatment [26]. As performance of

an algorithm trained on group-level data was found to be poor for each individual, a hybrid group- and individual-level approach to algorithm training and testing was subsequently used in the JITAI 'OnTrack'. The hybrid algorithm starts by learning from group-level data and is continuously updated with individual-level data from new app users to enable personalised dietary lapse prediction, also known as 'a warm start' approach [27,28]. A 'warm start' approach is commonly used by adaptive intervention designers, as it enables learning algorithms to make decisions when 'unknown states' are encountered [29]. Rather than guessing, information from the group is used to fill such gaps. However, this type of 'warm start' group- and individual-level approach has, to the best of our knowledge, not yet been applied to predict lapses within smoking cessation JITAIs.

As frequent EMAs can be burdensome for participants [30] and rely on consciously accessible rather than automatic processes, JITAI developers have begun to incorporate wearable sensor data streams to *detect* states of vulnerability and to trigger real-time support. With increased availability of consumer grade wearable sensors (e.g., smartwatches), physiological correlates of affect or self-regulatory capacity, such as steps or heart rate, can now be measured relatively unobtrusively in people's daily lives. Hybrid algorithms to *detect* acute stress and smoking behaviour from a suite of wearable sensors (i.e., chest- and wrist-worn sensors) have been used to identify decision points and tailoring variables in smoking cessation JITAIs [21,22,31–33]. However, additional variables beyond stress (e.g., cravings, cigarette availability) that could be easily self-reported and used to *predict* (rather than *detect*) lapses have to-date received less attention within these JITAIs. To the best of our knowledge, no available JITAI for smoking cessation has incorporated data streams from commercially and widely available smartwatches, such as a Fitbit device.

## The current smoking cessation JITAI project

As part of the current JITAI project, we recently developed and evaluated a series of group-level, individual-level, and hybrid supervised machine learning algorithms to *detect* (rather than *predict*) lapses in smokers attempting to stop, using routinely collected data from the popular Smoke Free app [34]. Similar to the abovementioned work by Goldstein and colleagues [26], we found that the group-level algorithm had variable performance when applied to new, unseen individuals. Separate algorithms trained and tested on each individual's data had improved performance but could only be produced for a minority of users due to a high proportion of non-lapsers in the dataset. As this initial study used an unprompted study design (i.e., users self-selected when they completed app diary entries, which may have biased users towards reporting non-lapses), we considered it important to triangulate the results with those from a prompted study design. The key contribution of the present study was therefore to add to the growing literature on the identification of JITAI decision points and tailoring variables by developing and evaluating a series of group-level, individual-level, and hybrid supervised machine learning algorithms to *predict* lapse risk in smokers attempting to stop, leveraging both EMAs and wearable sensor data. This was with a view to ultimately using the best-performing algorithm within a subsequent JITAI for smoking cessation.

Specifically, the present study aimed to collect prompted EMAs (e.g., cravings, stress, self-efficacy) and wearable sensor data (i.e., step count, heart rate) in adults attempting to stop smoking for 10 days, to address the following objectives:

1. To develop a series of group-level supervised machine learning algorithms and evaluate their ability to predict lapses for out-of-sample observations (i.e., randomly selected rows in the dataset).

2. To evaluate the ability of the best-performing group-level algorithm to predict lapses for out-of-sample individuals (i.e., group-to-individual generalisation).

3. To develop a series of individual-level supervised machine learning algorithms and evaluate their ability to predict lapses for out-of-sample individual observations.

4. To evaluate the ability of a hybrid (i.e., group- and individual-level) algorithm to predict lapses for out-of-sample individuals.

Across objectives 1–4, we aimed to develop and evaluate algorithms i) without and ii) with the wearable sensor data to examine their added value for lapse prediction. In addition, as an overarching objective, we aimed to select the overall best-performing and feasible algorithm to take forwards in a future JITAI (considered in the Discussion section).

## Methods

### Study design

This was a 10-day intensive, longitudinal, observational study with participants recruited between May 2022 and March 2023. Research shows that most smokers lapse and relapse (i.e., return to regular smoking) within the first week of the quit attempt [35]. We therefore opted for a high measurement frequency per day within this critical time window (i.e., 16 times per day) to capture rapid transitions from abstinence to lapse and relapse. The Checklist for Reporting EMA Studies (CREMAS) was used in the design and reporting of this study [36]. The study protocol and exploratory analysis plan were pre-registered on the Open Science Framework following piloting of the study protocol and materials but prior to data collection and analysis (https://osf.io/ywqpv). It is not standard practice to calculate *a priori* sample size requirements for the training and testing of supervised machine learning algorithms. We therefore conducted a simulation-based power analysis in R for a parallel, statistical model with assumptions informed by the available literature (for more details, see https://osf.io/ywqpv). It was estimated that a total of 40 participants and 16 survey prompts per day over a period of 10 days (i.e., 160 surveys per individual; 6400 total surveys) would provide >90% power (two-tailed alpha set to 5%) to detect a negative self-efficacy-lapse association, expressed as an odds ratio of 0.84.

### Eligibility criteria

**Inclusion criteria.** Smokers were eligible to participate if they: i) were aged 18+ years; ii) smoked cigarettes regularly; iii) resided in London and were willing to visit University College London (UCL) twice–i.e., before and after the 10-day study period; iv) owned a smartphone capable of running the required study smartphone apps (i.e., Android 8.0 or up; iOS 14.0 or up); v) were willing to set a quit date within 7 days from their initial study visit (preferably the next day, to capitalise on their motivation to stop); vi) were willing to wear a Fitbit and respond to hourly surveys for a period of 10 days during regular waking hours; vii) had internet/Wi-Fi access for the duration of the study; and viii) were able and willing to provide an exhaled carbon monoxide (eCO) measure (e.g., people with a diagnosis of asthma or COPD sometimes find it difficult to provide eCO measures).

**Exclusion criteria.** Smokers were not eligible to participate if they: i) had a known history of arrythmias (e.g., atrial fibrillation), ii) were regularly taking beta blockers (e.g., atenolol, bisoprolol), or iii) had an implanted cardiac rhythm device.

**Sample recruitment.** Participants were recruited through university mailing lists, leaflets on the university campus, word-of-mouth, and paid adverts on social media platforms (i.e., Facebook and Instagram).

**Payment to participants.** Participants were remunerated £50 for participation and were advised that for each day in the study that their EMA compliance was >70%, they would receive an additional £5 (i.e., they could earn up to £100). To receive the base rate of £50, participants were advised that they needed to complete ≥50% of the EMAs on ≥50% of the study days. However, participants who did not meet the minimum response rate were given £20 as compensation for their time and travel. It was explained that remuneration was not linked to participants' smoking behaviour. Participants also received free access to the 'pro' (paid) version of the Smoke Free app (https://smokefreeapp.com/). Smoke Free provides evidence-informed behavioural support, has a large user base (~3,000 new downloads per day). In a large randomised controlled trial among motivated smokers provided with very brief advice to quit, the offer of the Smoke Free app did not have a detectable benefit for cessation compared with follow-up only. However, the app increased quit rates when smokers randomised to receive the app downloaded it [37].

**Ethics.** Ethical approval was obtained from the UCL Research Ethics Committee (Project ID: 15297.004). Participants were asked for their informed consent to share their anonymised research data with other researchers via an open science platform.

**Measures and procedure.** Interested participants were asked to complete an online screening survey to determine eligibility and describe the sample (see the Supporting Information, S1 Table). Participants were asked to provide information about demographic (e.g., age, gender, ethnicity, education) and smoking (e.g., cigarettes per day, time to first cigarette, motivation to stop) characteristics (see the Supporting Information, S2 Table).

**Onboarding visit.** Eligible participants were invited to attend an in-person visit at UCL to learn more about the study devices and procedures. They were loaned a Fitbit Charge 4 [38,39] and asked to download the m-Path app [40]. Participants received instructions on how to use the m-Path app and the Fitbit (including when to charge it). We specifically selected a commercially available and research validated wearable sensor that is widely available and integrated with smartphones [41,42]. Participants were asked to set a quit date within the next 7 days (preferably the next day to capitalise on their motivation to stop) and were advised to download and use the Smoke Free app to aid their quit attempt. Participants were asked if they wanted to add up to three participant-specific factors to be prompted about (i.e., an open-ended question encouraging participants to list any other factor that might influence their lapse risk, which was not already covered in the measurement battery). During the onboarding visit, it was emphasised that it was important for participants to continue to respond to the hourly surveys also when they had smoked.

**10-day intensive, longitudinal study.** The 10-day study period started on each participant's quit date. During the 10-day study period, participants received 16 hourly prompts per day via the m-Path app (i.e., signal-contingent sampling), scheduled within their usual waking hours. For example, a participant who indicated that they typically wake up at 7am received hourly prompts from 7am to 10pm. Each survey took 1–3 minutes to complete. Responses were time stamped and interactions needed to be completed within 30 minutes. Participants were also asked to record lapses as and when they occurred (i.e., event-contingent sampling) via an in-app button. To promote EMA adherence, participants were sent regular e-mails from one of the researchers with updates on their adherence rate and graphs of their survey responses (e.g., cravings, affect), which were automatically generated by the m-Path software.

**Follow-up visit.** Immediately after the 10-day study period, participants were invited to attend an in-person follow-up visit at UCL to measure their eCO using a Bedfont iCOquit Smokerlyzer (https://resources.bedfont.com/wp-content/uploads/2024/03/LAB806-iCOquit-manual-issue-13.pdf) to verify abstinence (the cut-off was set to 8ppm), return the Fitbit, deactivate the study-related apps (e.g., export and delete the data from the Fitbit and m-Path apps), and receive payment for participation.

**Outcome (or target) variable.** The outcome variable was whether participants reported having lapsed (no vs. yes) in the last hour or pressed the button to indicate that they had lapsed. In sensitivity analyses, the outcome variable was cravings, measured on an 11-point Likert scale and dichotomised into low (0–6) vs. high (7–10) cravings.

**Explanatory (or input) variables.** At signal- and event-contingent survey prompts, participants were asked to provide information on positive affect (e.g., happy, enthusiastic) and negative affect (e.g., sad, stressed)–informed by the circumplex model of affect [43–45]–in addition to information on cravings, confidence in their ability to stay quit, motivation to stay quit, bodily pain, social context, physical context, cigarette availability, alcohol consumption, caffeine consumption, nicotine use, and up to three participant-specific factors selected during the onboarding visit (see the Supporting Information, S3 Table, for the EMA items and response options). As most participants who added their own survey item repeated things already included in the survey but phrased slightly differently (e.g., 'presence of other smokers'), we did not consider the participant-specific factors further in the current analyses. Where appropriate, 11-point Likert scales were used. We consulted the literature for previously used/validated EMA items [46–49].

Data were collected via the Fitbit Charge 4 on participants' heart rate and step count. As heart rate is acutely influenced by factors such as caffeine consumption [50], posture (e.g., sitting, standing, walking), smoking abstinence, and nicotine use [51], participants were asked to provide information about these factors as part of the hourly EMAs. We had planned to examine heart rate variability in our analyses; however, following discussion with sensor data experts and due to the insufficient Fitbit Charge 4 sampling rate (<1 Hz, with sampling frequencies of 250–1000 Hz recommended for heart rate variability analyses), we opted not to include heart rate variability in our analyses and focused instead on heart rate.

We had also planned to collect data on participants' engagement with the Smoke Free app (i.e., logins, time spent per login); however, due to limited resource, engagement metrics were not considered in the current analyses.

## Data analysis

All analyses were conducted using the R Statistical Software [52]. First, descriptive statistics (e.g., the mean and standard deviation) were calculated to describe the characteristics of the overall sample, the analytic sample, and those excluded due to not meeting the adherence cut-offs. T-tests and Chi-squared tests, as appropriate, were used to compare the characteristics of the analytic sample with those excluded.

Next, the main analyses used the *tidymodels* framework of packages [53], setting the engine to the relevant algorithm (e.g., *ranger* for Random Forest or *glmnet* for Penalised Logistic Regression) and the mode to *classification*. Four different types of supervised machine learning algorithms were trained and tested: Random Forest (RF), Support Vector Machine (SVM), Penalized Logistic Regression, and Extreme Gradient Boosting (XGBoost), selected based on their relatively low computational demands (as the algorithm will ultimately be implemented within a smartphone app or similar technology), the availability of off-the-shelf R packages, and their relatively good interpretability compared with approaches such as deep learning (see Perski, Li et al., 2023 for an overview of the algorithms). As we aimed to *predict* rather than *detect* lapses, the input variables were used to predict the target variable (i.e., lapse incidence) at the subsequent time point (i.e., the next hour).

We had also planned to run a series of generalised linear mixed models in parallel to assess between- and within-person predictor-lapse associations (typically used to analyse EMA data); however, as this type of statistical model addresses different research questions, we did not run such additional analyses here.

## Data pre-processing steps

Due to uneven Fitbit sampling or storage (outside of the researchers' control), we first rounded the heart rate data to 5-second intervals. The data frame was then expanded to all possible 5-second intervals during the participant-specific time window (16 hours on each of the 10 study days). This was repeated for the step count data, which was rounded to 1-minute intervals (due to the Fitbit Charge 4 sampling or storage rate).

Next, participants with insufficient EMA adherence were removed from the analytical sample. Different adherence cut-offs were explored ($\geq$50%, $\geq$60%, $\geq$70%), with $\geq$60% selected as this maximised data completeness and the number of participants that could be retained for analysis. Similar to Chevance and colleagues, missing EMAs within the analytical sample were imputed with the univariable Kalman filter [54].

Finally, participants with insufficient sensor wear time were removed from the analytical sample. Participants with sufficient wear time were defined as having $\geq$20% adherence on $\geq$5 of the 10 study days. In preparation for the feature extraction, different prediction distances and time windows were set up. In line with the exploratory nature of the analyses incorporating the sensor data, we know little about the optimal temporal distance from the EMA prompt to use ('prediction distance'; i.e., 15 minutes, 30 minutes, or 45 minutes from the EMA prompt) and the amount of data to include when deriving sensor data features of interest ('time window'; 5 minutes, 10 minutes, or 15 minutes of data). Following Bae and colleagues, in a series of unplanned analyses, we explored different combinations of prediction distances and time windows [55]. Missing heart rate and step data within the analytical sample were imputed for each combination of prediction distance and time window (e.g., 15-minute distance from the EMA prompt incorporating 5 minutes of sensor data) with the univariable Kalman filter. Visual inspection was used to assess the plausibility of the imputed time series.

## Feature extraction

Relevant features were extracted from the sensor data, including the standard deviation within the given prediction distance and time window, the change in slope (estimated using a linear model), the minimum and maximum value within the interval, and the maximum rate of change within the interval. As above, given the dearth of prior studies harnessing sensor data in the prediction of momentary smoking lapse risk, the specific features extracted and tested were exploratory in nature and shaped by conversations with sensor data experts.

## Model training and testing

First, we created train-test splits of the datasets, with 80% of the data used for training and 20% kept for testing (i.e., an unseen 'holdout sample'). Next, the models were trained using $k$-fold cross-validation [56], with $k$ set to 10. The models were subsequently tested on the unseen hold-out samples and performance metrics were calculated. This approach was applied to the group- and individual-level algorithms (see below for more details about the hybrid algorithms). The analyses proceeded as follows: objectives 1–4 were first addressed through training and testing algorithms using the EMA data only. Next, the analyses were repeated using the EMA and wearable sensor data. Finally, sensitivity analyses were performed using the EMA data only but with cravings as the outcome variable of interest (dichotomised into 'low' and 'high' craving scores for ease of interpretation). The sensitivity analyses were planned due to the observation in our previous study that individual-level and hybrid algorithms could not be constructed for all participants due to some reporting either 0% or 100% lapses [34]. Although a focus on cravings as outcome would imply a slightly different theoretical model (as this would assume that it is important to intervene earlier in the chain of events, focusing on

factors that predict cravings rather than those that predict lapses), we were interested in whether such a focus would increase the number of participants for whom individual-level and hybrid algorithms could be constructed. The sensitivity analyses are reported in the Supporting Information, S7–S11 Figs.

Predicted and observed outcomes were compared to estimate model accuracy (i.e., the proportion of true positives and true negatives), sensitivity (i.e., the true positive rate) and specificity (i.e., the true negative rate). Estimates were compared with pre-specified thresholds for acceptable accuracy (.70), sensitivity (.70) and specificity (.50) [57]. It is more costly for a future JITAI to miss a true positive (lapse) than a true negative (non-lapse) because the former may set the individual on a trajectory towards full relapse, and with support provided in lower-risk situations unlikely to be harmful to individuals, which explains our lower specificity threshold (.50). In addition, algorithm performance was evaluated by calculating an area under the receiver operating characteristic curve (AUC) estimate and an accompanying 95% confidence interval (CI) using the *pROC* package [58]. The AUC captures the trade-off between sensitivity and specificity. AUC estimates with CIs that include .50 (i.e., chance performance) were considered unacceptable.

## Objective 1—Identifying a best-performing group-level algorithm

An optimal group-level algorithm (e.g., Random Forest, Support Vector Machine, Penalised Logistic Regression, XGBoost) was identified, defined as the algorithm that most closely met the pre-specified thresholds for acceptable accuracy, sensitivity, and specificity. In the event of multiple algorithms meeting the thresholds, the algorithm with the greatest AUC value was selected. Each algorithm requires specific hyperparameters (e.g., penalty, cost) to be set. As is common within the machine learning field, model-specific hyperparameters were 'tuned'–i.e., identified in a data-driven manner through systematically evaluating predictions from algorithms with different candidate configurations across a given hyperparameter search space–to minimise the generalisation error and optimise algorithm performance [59]. To limit the computational demands for the algorithms with a large number of hyperparameters (i.e., XGBoost), a technique called 'space-filling' was used to construct manageable hyperparameter search spaces using the Latin hypercube design, which constructs parameter grids that attempt to cover the entire parameter space but without testing every possible configuration [60]. For the best-performing group-level algorithm, the *vip* package [61] was used to estimate the permutation-based, model-agnostic feature importance (i.e., the most influential predictor variables).

## Objective 2—Performance of the best-fitting group-level algorithms for out-of-sample individuals

Leave-one-out cross-validation was used to examine the performance of the best-performing group-level algorithm identified as part of Objective 1 for 'unseen' individuals, who were each omitted from the training set and used for testing (i.e., the procedure was repeated for each individual in the dataset). Participants with 0% or 100% lapses were excluded. Performance of the best-performing group-level algorithm was deemed acceptable if the median AUC across individuals was greater than chance performance (i.e., 0.5).

## Objective 3—Identifying best-performing individual-level algorithms

Next, algorithms were separately trained and tested on each individual's data. We took the same approach as that used in [34] and set the cut-off for inclusion in the individual-level analyses to participants with >5 reported lapses and >5 reported non-lapses. The permutation-

based, model-agnostic feature importance (i.e., the most influential predictor variables) was estimated for each individual.

## Objective 4—Performance of a hybrid model for individuals

Next, the analyses conducted as part of Objective 2 were repeated, with 20% of each individual's data included in the training sets. The remaining 80% of the individual's data was used for testing. Performance was deemed acceptable if the median AUC across individuals was greater than chance performance (i.e., 0.5). We also compared the median AUC with that produced as part of Objective 2.

## Overarching objective–Selecting the overall best-performing and feasible algorithm to take forwards to underpin a JITAI

Finally, the algorithm that most closely met the pre-specified thresholds across individuals was identified. Selection was based on the median AUC across individuals. This deviated slightly from the pre-registered study protocol, in which we had suggested to look across three performance indicators (i.e., accuracy, sensitivity, and specificity). In addition, as the implementation of the selected algorithm within a JITAI comes with additional considerations, we also assessed i) theoretical clarity; ii) acceptability for participants; and iii) scalability and technical feasibility for intervention developers. These additional considerations had not been specified in the pre-registered study protocol.

## Results

A total of 147 participants were eligible for inclusion, of whom 46 participated in the study (46/147; 31.3%). Of the 46 participants, 38 met the EMA adherence cut-off (38/46; 82.6%) and were included in the initial set of analyses (see Fig 1). Of the 38 participants who met the EMA adherence cut-off, 30 met the sensor data cut-off (30/38; 78.9%) and were included in the second set of analyses.

Participants ($n$ = 38) were predominantly female (60.5%), aged an average of 42.9 (SD = 14.3) years, and most had post-16 educational qualifications (86.8%; see Table 2). Participants smoked an average of 13.0 (SD = 6.4) cigarettes per day.

The average percentage EMA adherence was 76.9% (SD = 9.2%). Participants responded to a total of 6,124 (6,080 signal-contingent and 44 event-contingent) EMAs. The proportion of lapses (vs. non-lapses) reported across the EMAs was 6.9% (423/6,124); however, this varied widely across participants, with a median of 1.6% lapses (range: 0%-74.1%). A total of 13 participants (13/38; 34.2%) reported 0% lapses during the 10-day study period, with 25 participants (25/38; 65.8%) reporting that they had smoked since the quit date when they attended the in-person follow-up assessment.

### EMA data only

**Objective 1—Identifying a best-performing group-level algorithm.** The best-performing group-level algorithm was a Random Forest (RF) algorithm (AUC = 0.899, 95% CI = 0.871 to 0.928; see Fig 2). This was closely followed by an Extreme Gradient Boosting (XGBoost) algorithm, with an AUC of 0.895 (95% CI = 0.862 to 0.928), a Support Vector Machine (SVM) algorithm (AUC = 0.886, 95% CI = 0.853 to 0.918), and a Penalised Logistic Regression algorithm (AUC = 0.879, 95% CI = 0.846 to 0.911). See the Supporting Information, S1 Fig for a visual comparison of the AUCs. The variable importance plots for the best-performing group-level algorithms are presented in the Supporting Information, S2 Fig.

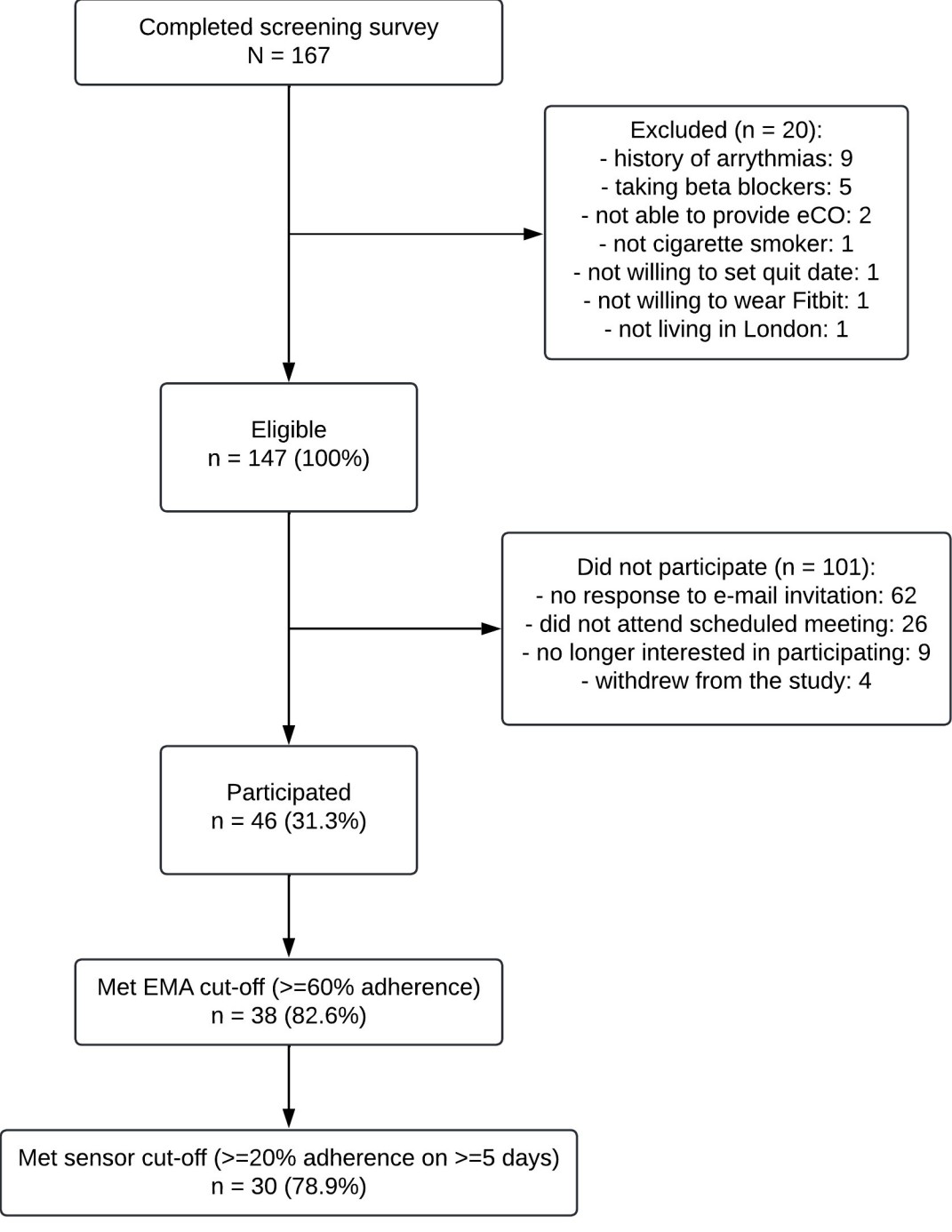

**Fig 1. Participant flow chart illustrating the participant screening, enrolment, and inclusion in the study analyses.**

The most influential predictor variables for the best-performing group-level RF algorithm included confidence (time-varying), motivation to stay quit (time-varying), age (time-invariant), whether the prior event was a lapse (time-varying), and whether cigarettes were available (time-varying; see Fig 3).

**Objective 2—Performance of the best-fitting group-level algorithms for out-of-sample individuals.** After removing the 13 participants with 0% lapse events, algorithm performance

**Table 2. Participant demographic and smoking characteristics.**

| Characteristic | Overall, N = 46 | Excluded, N = 8 | Included, N = 38 | p-value[2] |
|---|---|---|---|---|
| Age[1] | 44.22 (13.56) | 50.50 (7.27) | 42.89 (14.26) | 0.04 |
| Gender | | | | 0.2 |
| Female | 26 (56.5%) | 3 (37.5%) | 23 (60.5%) | |
| Male | 20 (43.5%) | 5 (62.5%) | 15 (39.5%) | |
| Cigarettes per day[1] | 13.43 (6.57) | 15.50 (7.31) | 13.00 (6.43) | 0.4 |
| Occupation | | | | 0.6 |
| Non-manual | 20 (43.5%) | 3 (37.5%) | 17 (44.7%) | |
| Manual | 10 (21.7%) | 1 (12.5%) | 9 (23.7%) | |
| Other (e.g., student, unemployed, retired) | 16 (34.8%) | 4 (50.0%) | 12 (31.6%) | |
| Post-16 educational qualifications | 40 (87.0%) | 7 (87.5%) | 33 (86.8%) | >0.9 |
| Ethnicity | | | | >0.9 |
| Asian or Asian British (any Asian background) | 4 (8.7%) | 1 (12.5%) | 3 (7.9%) | |
| Black, Black British, Caribbean or African (any Black, Black British or Caribbean background) | 1 (2.2%) | 0 (0.0%) | 1 (2.6%) | |
| Mixed or multiple ethnic groups (e.g., White and Black African, White and Asian) | 2 (4.3%) | 0 (0.0%) | 2 (5.3%) | |
| Other ethnic group | 1 (2.2%) | 0 (0.0%) | 1 (2.6%) | |
| White (any White background) | 38 (82.6%) | 7 (87.5%) | 31 (81.6%) | |
| Time to first cigarette | | | | 0.3 |
| Within 5 minutes | 12 (26.1%) | 3 (37.5%) | 9 (23.7%) | |
| 6–30 minutes | 21 (45.7%) | 5 (62.5%) | 16 (42.1%) | |
| 31–60 minutes | 6 (13.0%) | 0 (0.0%) | 6 (15.8%) | |
| After 60 minutes | 7 (15.2%) | 0 (0.0%) | 7 (18.4%) | |
| Motivation to stop | | | | 0.2 |
| I don't want to stop smoking | 0 (0.0%) | 0 (0.0%) | 0 (0.0%) | |
| I think I should stop smoking but don't really want to | 2 (4.3%) | 0 (0.0%) | 2 (5.3%) | |
| I want to stop smoking but haven't thought about when | 6 (13.0%) | 0 (0.0%) | 6 (15.8%) | |
| I really want to stop smoking but don't know when I will | 7 (15.2%) | 0 (0.0%) | 7 (18.4%) | |
| I want to stop smoking and hope to soon | 12 (26.1%) | 3 (37.5%) | 9 (23.7%) | |
| I really want to stop smoking and intend to in the next 3 months | 4 (8.7%) | 2 (25.0%) | 2 (5.3%) | |
| I really want to stop smoking and intend to in the next month | 15 (32.6%) | 3 (37.5%) | 12 (31.6%) | |
| Past-year quit attempt | | | | 0.5 |
| No, never | 3 (6.5%) | 0 (0.0%) | 3 (7.9%) | |
| Yes, but not in the past year | 21 (45.7%) | 5 (62.5%) | 16 (42.1%) | |
| Yes, in the past year | 22 (47.8%) | 3 (37.5%) | 19 (50.0%) | |
| Smoked since quit date (assessed at follow-up, with abstinence eCO-verified) | 32 (69.6%) | 7 (87.5%) | 25 (65.8%) | 0.2 |
| % Completed EMAs[1] | 69.04 (19.97) | 31.48 (13.05) | 76.94 (9.16) | <0.001 |
| Ever use of pharmacological support (e.g., NRT, varenicline, e-cigarettes) | 38 (82.6%) | 7 (87.5%) | 31 (81.6%) | 0.7 |
| Ever use of behavioural support (e.g., counselling, Quitline, website, app) | 19 (41.3%) | 3 (37.5%) | 16 (42.1%) | 0.8 |

[1]Mean (SD); n (%)

[2]Welch Two Sample t-test; Pearson's Chi-squared test

could be computed for 25 participants (25/38; 65.8%). The median AUC was moderate at 0.639 and this metric varied widely across participants (range: 0.524–0.994).

**Objective 3—Identifying best-performing individual-level algorithms.** After removing participants with insufficient lapse and non-lapse events, algorithm performance could be computed for 15 participants (15/38; 39%). Fig 4 illustrates the frequency distribution of the

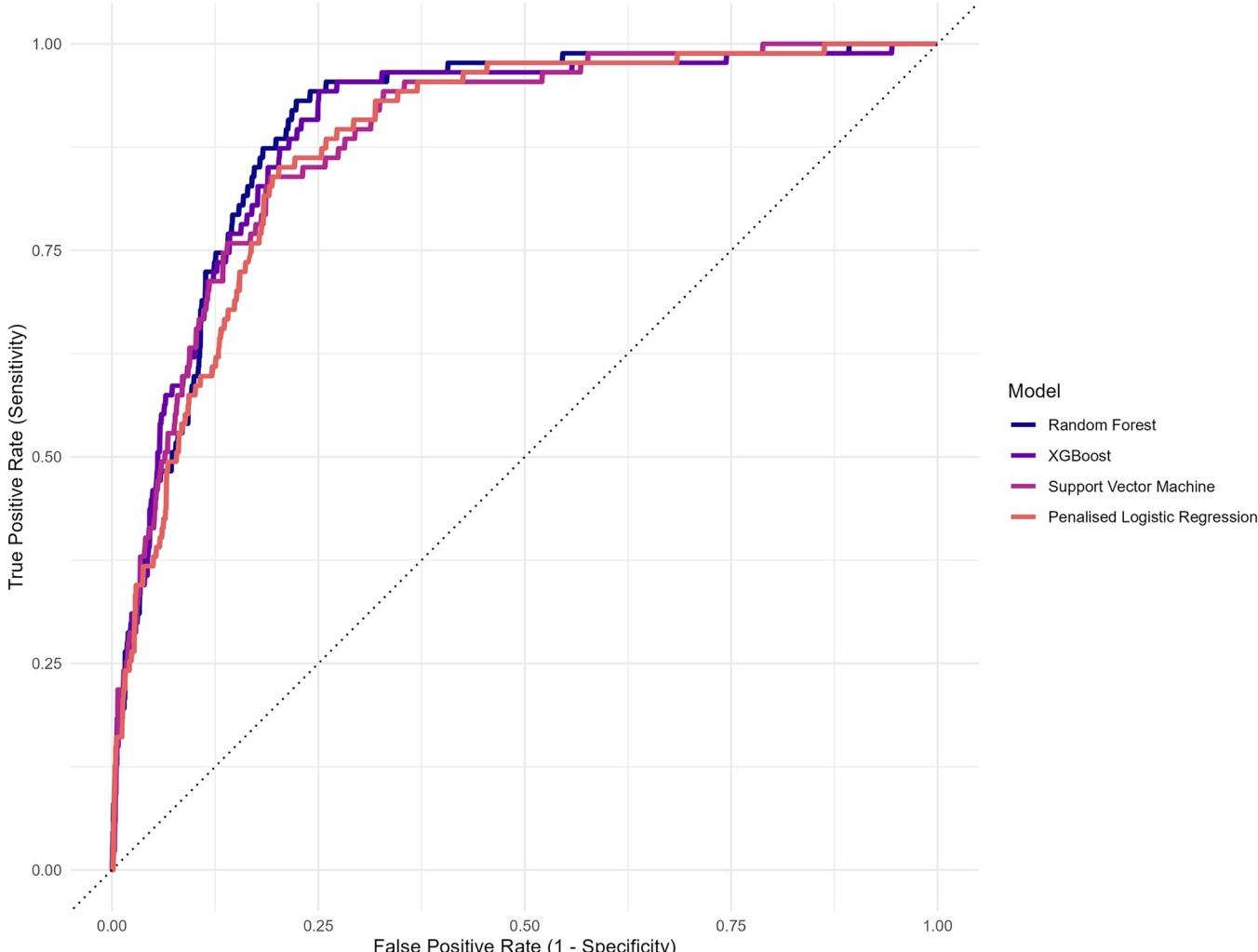

**Fig 2. Plot of the area under the receiver operating characteristic curve (AUC) estimate for each of the group-level algorithms (without sensor data).**

performance metrics of interest for participants' best-performing algorithms. The median AUC for participants' best-performing algorithms was 0.855 (range: 0.451 to 1.000).

In an analysis examining the number of participants for whom the individual-level algorithm provided a benefit over the group-level algorithm (based on the AUC), the individual-level algorithm was superior for just over half of the participants (8/15; 53.3%).

Next, we examined the proportion of participants with each of the predictor variables in their top 10 list, estimated using the *vip* function applied to their best-performing individual-level algorithm (n = 15; see the Supporting Information, S3 Fig). For example, 'activity–eating' and 'excitement' were included in 40% of participants' top 10 lists.

**Objective 4—Performance of a hybrid model for individuals.**   When repeating the analyses conducted to address Objective 2 but with 20% of the individual's data included in the training set (n = 25), the median AUC was 0.692 (range: 0.523 to 0.998). The hybrid algorithm could be produced for 25 individuals compared with the 15 for whom individual-only algorithms could be produced. The hybrid algorithm was superior to the group-level algorithm for 48.0% (12/25) of participants (based on the AUC).

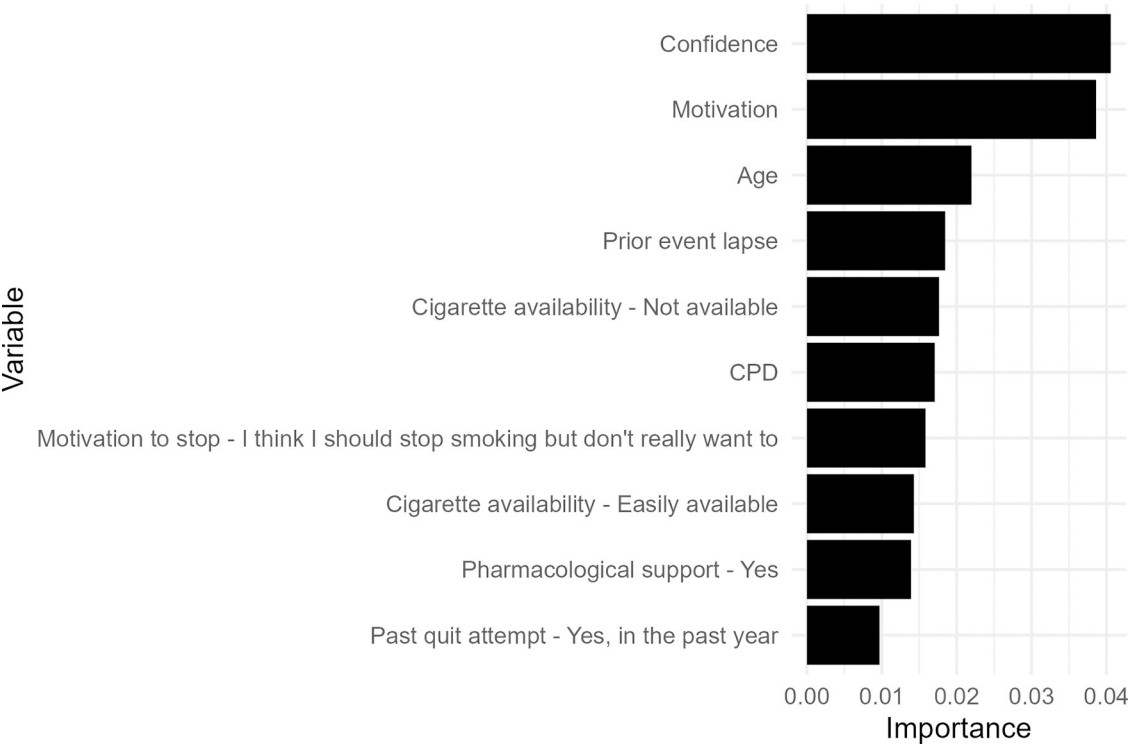

**Fig 3. Variable importance plot for the best-performing group-level Random Forest algorithm. The variable importance score does not indicate the direction of the relationship between the predictor and outcome variable.**

### EMA and sensor data

**Objective 1—Identifying a best-performing group-level algorithm.** First, the best prediction distance-time window combination was examined through training and testing each of the four algorithm types using each of the prediction distance-time window combinations (i.e., 4 algorithm types * 9 combinations = 36 models). This resulted in the selection of prediction distance 1 (15 minutes prior to the EMA prompt) combined with time window 3 (15 minutes of data; see the Supporting Information, S4 Table). All subsequent analyses used this prediction distance-time window combination.

The best-performing group-level algorithm was an RF algorithm (AUC = 0.952, 95% CI = 0.933 to 0.970; see Fig 5). This was closely followed by a Penalised Logistic Regression algorithm (AUC = 0.944, 95% CI = 0.921 to 0.966), an Extreme Gradient Boosting (XGBoost) algorithm (AUC = 0.933, 95% CI = 0.907 to 0.959), and a Support Vector Machine (SVM) algorithm (AUC = 0.865, 95% CI = 0.822 to 0.909). See the Supporting Information, S4 Fig for a visual comparison of the AUCs. The variable importance plots for the best-performing group-level algorithms are presented in the Supporting Information, S5 Fig.

The most influential predictor variables for the best-performing group-level RF algorithm included motivation (time-varying), confidence (time-varying), whether the prior event was a lapse (time-varying), whether cigarettes were available (time-varying), and age (time-invariant; see Fig 6).

**Objective 2—Performance of the best-fitting group-level algorithms for out-of-sample individuals.** After removing participants with 0% lapses, algorithm performance could be computed for 20 participants (20/30; 66.7%). The median AUC was moderate at 0.745 and varied widely across participants (range: 0.494–0.979).

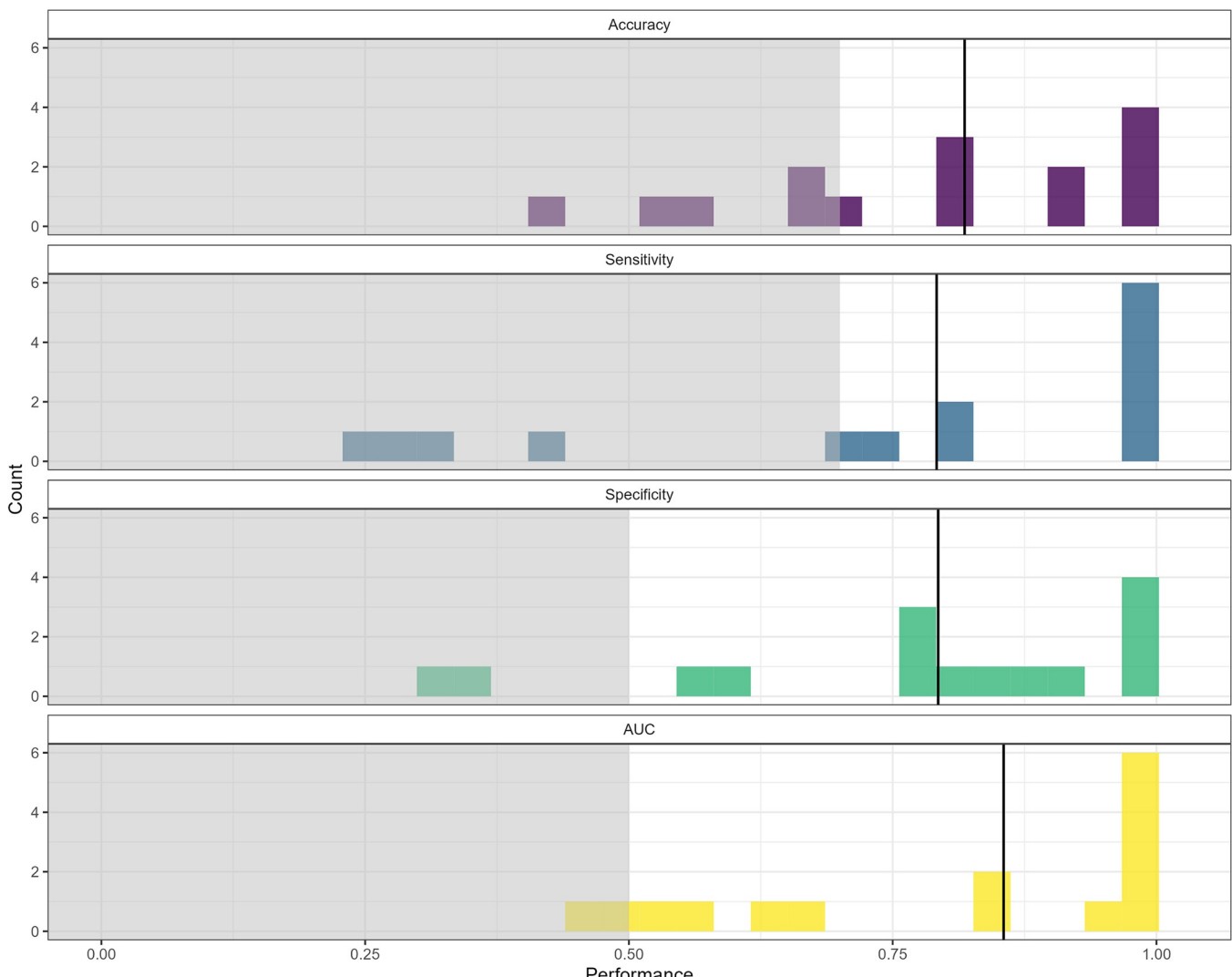

**Fig 4. Frequency distributions of the performance metrics of interest (i.e., accuracy, sensitivity, specificity, AUC) for the best-performing individual-level algorithms (n = 15).** The shaded grey areas represent the prespecified thresholds for acceptable accuracy (0.70), sensitivity (0.70), specificity (0.50), and AUC (0.50). The solid vertical lines represent the median.

**Objective 3—Identifying best-performing individual-level algorithms.** After removing participants with insufficient lapse and non-lapse events, algorithm performance metrics could be computed for 11 participants (11/30; 37%). Fig 7 illustrates the frequency distribution of the performance metrics of interest for participants' best-performing algorithms. The median AUC for participants' best-performing algorithms was 0.983 (range: 0.549 to 1.000).

In an analysis examining the number of participants for whom the individual-level algorithm provided a benefit over the group-level algorithm (based on the AUC), the individual-level algorithm was superior for most participants (9/11; 81.8%).

Next, we examined the proportion of participants with each of the predictor variables in their top 10 list, estimated using the *vip* function applied to their best-performing individual-level algorithm (n = 11; see the Supporting Information, S6 Fig). For example, 'change in slope–heart rate' and 'confidence' were included in 50% and 40% of participants' top 10 lists, respectively.

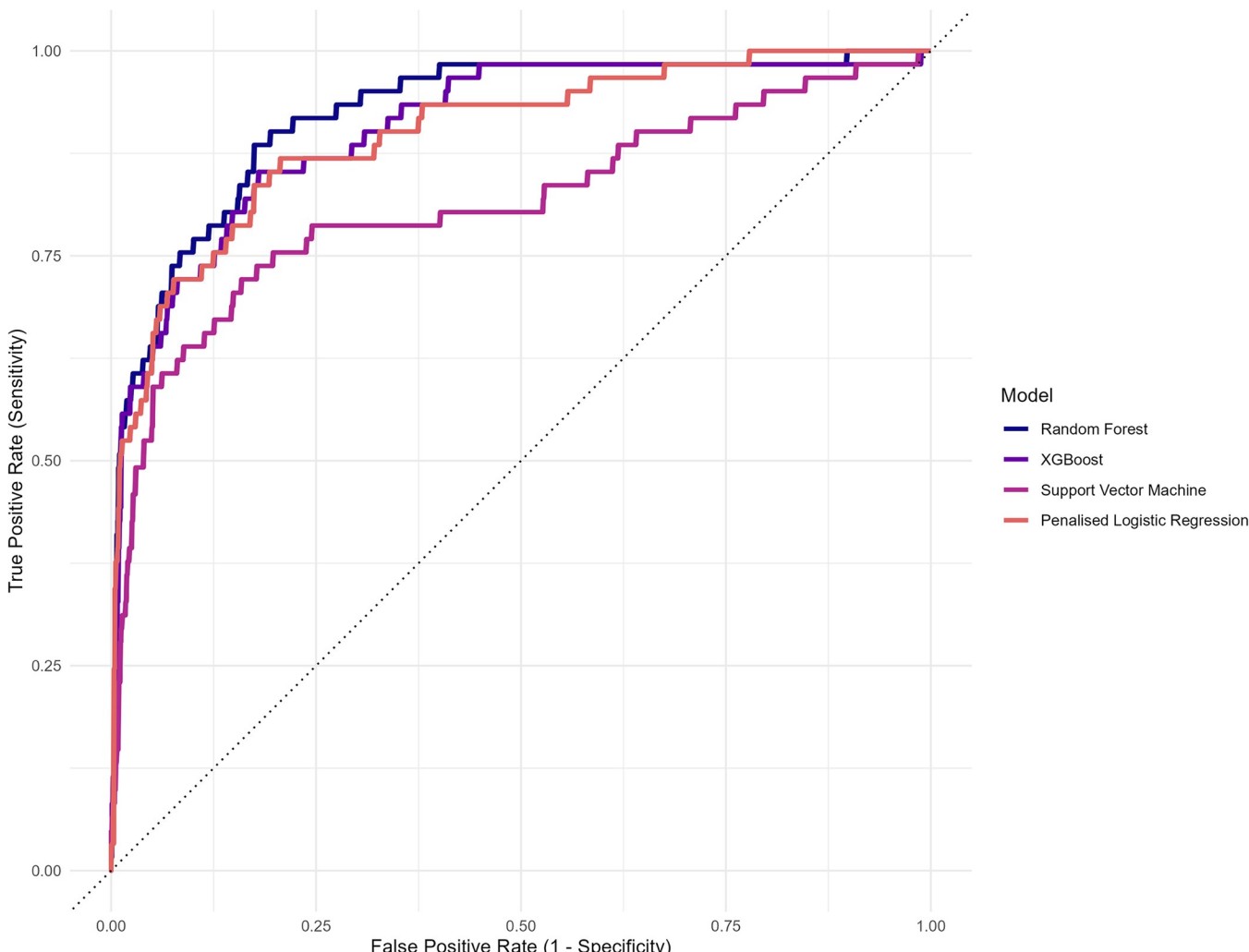

**Fig 5. Plot of the area under the receiver operating characteristic curve (AUC) estimate for each of the group-level algorithms (with sensor data).**

**Objective 4—Performance of a hybrid model for individuals.** When repeating the analyses conducted to address Objective 2 but with 20% of the individual's data included in the training set (n = 20), the median AUC was 0.772 (range: 0.444 to 0.968). The hybrid algorithm could be produced for 20 individuals compared with the 11 for whom individual-only algorithms could be produced. The hybrid algorithm was superior to the group-level algorithm for 55% (11/20) of participants (based on the AUC).

## Planned and unplanned sensitivity analyses

**EMA data only, with cravings as outcome.** The planned sensitivity analyses with craving scores ('high' versus 'low') as outcome are reported in the Supporting Information, S7–S11 Figs. The results remained largely robust (e.g., the best-performing group-level algorithm was an RF algorithm); however, individual- and hybrid-level algorithms could be constructed for a greater number of participants (i.e., 31 and 37 participants, respectively).

**EMA data only, training and testing an artificial neural net.** As a result of the review process, we subsequently compared the best-performing group-level algorithm (i.e., Random

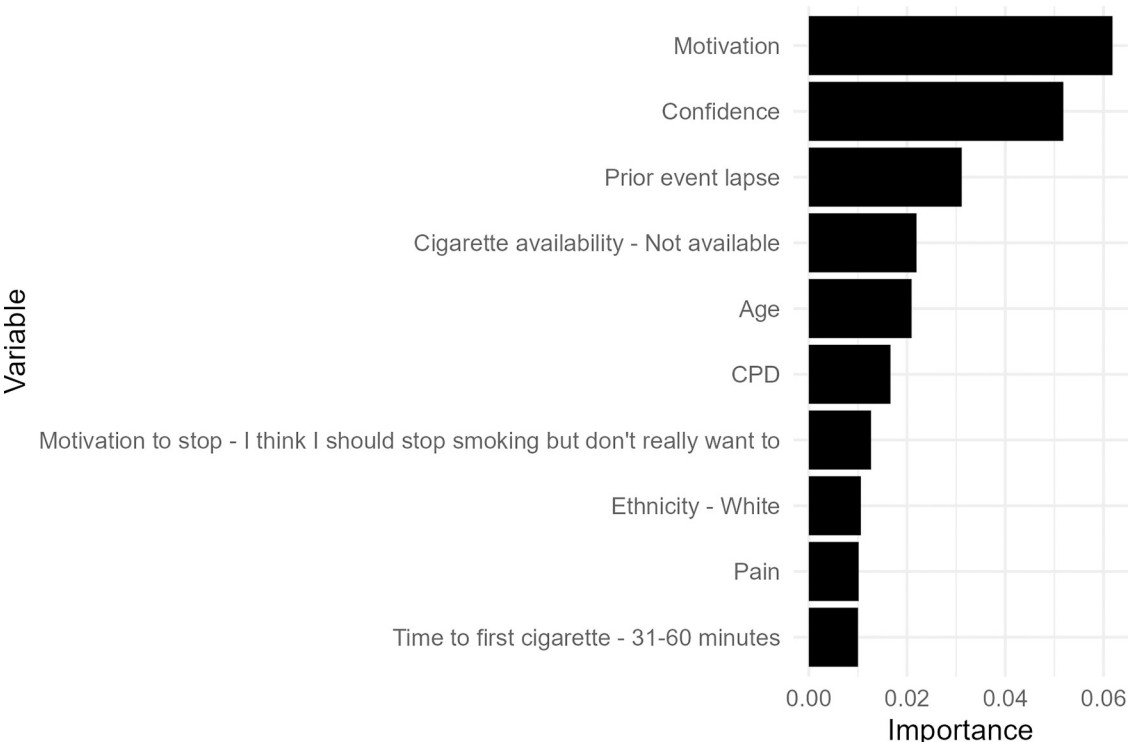

**Fig 6. Variable importance plot for the best-performing group-level Random Forest algorithm (with sensor data).** The variable importance score does not indicate the direction of the relationship between the predictor and outcome variable.

Forest) with a simple Artificial Neural Network. Similar to the other algorithm types, the *tidymodels* framework of packages [53] was used, setting the engine to *nnet* (i.e., a single-layer, feed-forward neural net), the mode to *classification*, and tuning the relevant hyperparameters. The algorithm performance remained similar (but was not superior to) the best-performing group-level algorithm (AUC = 0.853, 95% CI = 0.818–0.887).

After removing participants with 0% lapses, algorithm performance could be computed for 25 participants (25/30; 65.8%). The median AUC was moderate at 0.614 and varied widely across participants (range: 0.509–0.954). Due to non-superiority, we did not proceed with the remaining objectives.

**EMA data only, incorporating a seasonality indicator.**   As a result of the review process, we also examined whether an indicator of seasonality (i.e., the time of year when the participants were recruited into the study) influenced algorithm performance. We incorporated the season of the year as a categorical predictor (i.e., summer, autumn, winter, spring). The algorithm type (i.e., Random Forest) and performance (AUC = 0.912, 95% CI = 0.887–0.937) remained similar (but was not superior to) the best-performing group-level algorithm without the inclusion of seasonality.

After removing participants with 0% lapses, algorithm performance could be computed for 25 participants (25/30; 65.8%). The median AUC was moderate at 0.664 and varied widely across participants (range: 0.511–0.981). Due to non-superiority, we did not proceed with the remaining objectives.

**EMA and sensor data, removing timepoints with potential confounders.**   Additional unplanned sensitivity analyses with timepoints with potential confounders removed (i.e., when responses indicated walking/exercising, caffeine intake, nicotine use) are reported in the

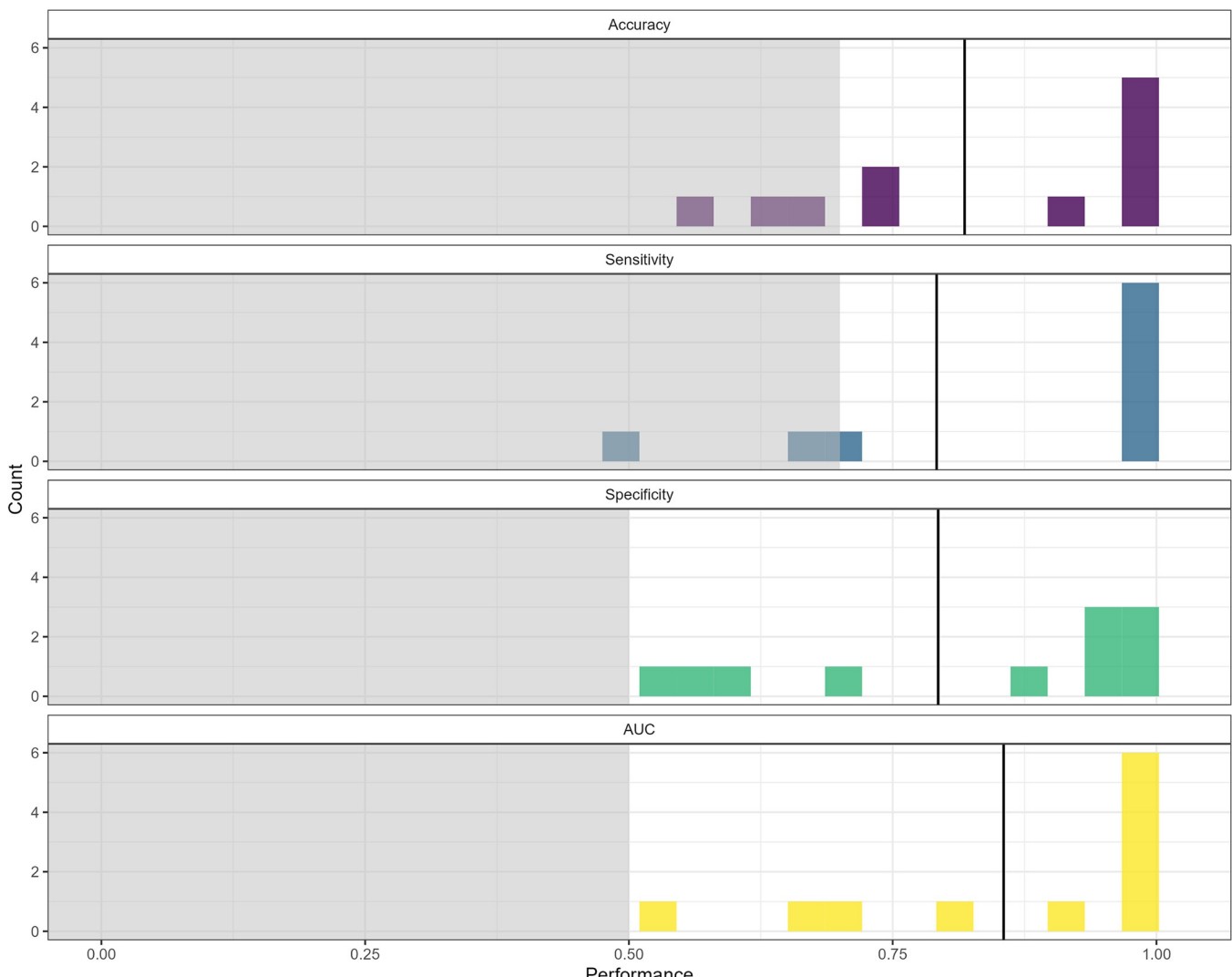

**Fig 7. Frequency distributions of the performance metrics of interest (i.e., accuracy, sensitivity, specificity, AUC) for the best-performing individual-level algorithms (n = 11).** The shaded grey areas represent the prespecified thresholds for acceptable accuracy (0.70), sensitivity (0.70), specificity (0.50), and AUC (0.50). The solid vertical lines represent the median.

Supporting Information, S12–S13 Figs. The results remained largely robust; however, individual- and hybrid-level algorithms could be constructed for fewer participants (i.e., 9 and 17 participants, respectively).

## Discussion

To inform the development of a future JITAI for lapse prevention, EMAs and wearable sensor data were used to train and test a series of supervised machine learning algorithms to distinguish lapse from non-lapse events. We found that high-performing group-level lapse prediction algorithms without and with passively sensed data had variable performance when applied to unseen individuals. Individual-level and hybrid algorithms had improved performance, particularly when incorporating sensor data from a commercially available device for participants with sufficient wear time. However, they could only be constructed for a limited number of individuals due to methodological challenges (discussed in detail below). Hybrid

algorithms could be constructed for about twice as many individuals as the individual-level algorithms. Overall, Random Forest algorithms outperformed the other algorithm types (i.e., Support Vector Machine, XGBoost, Penalised Logistic Regression); however, it should be noted that the absolute AUC differences were minimal. The Random Forest algorithm has gained popularity due to its flexibility with regards to the functional form of predictor-outcome associations and ability to account for many predictor variables (including higher-order interactions) [62]. Some studies show that the Random Forest algorithm outperforms similar algorithms, such as the Support Vector Machine, although results are mixed [62]. We therefore caution against placing too much emphasis on the algorithm type in the present study. Using the variable importance method, predictors such as feeling confident in one's ability to stay quit and feeling motivated to stay quit appeared important for lapse prediction at the group-level. As expected, however, the individual-level analyses indicated that different predictors were important for different individuals. These results can be harnessed by JITAI developers to identify plausible intervention options to try at times of predicted need. However, as highlighted in our previous work [34], we caution against overreliance on the results from the variable importance function and recommend that robustness analyses (e.g., partial dependence plots) and external validation are carried out prior to determining which predictors are most important.

## Selecting the overall best-performing and feasible algorithm to take forwards to underpin a JITAI

Multiple success criteria (e.g., acceptability, scalability, technical feasibility) need to be carefully balanced against algorithm performance in the JITAI development and implementation process. Below, we discuss such considerations in the light of our results.

### Theoretical clarity

The use of craving scores as the outcome (as opposed to lapse incidence) increased the number of participants for whom individual-level and hybrid algorithms could be constructed. However, a focus on cravings rather than lapses would imply a slightly different theoretical model, assuming that it is most important to intervene on the precursors of cravings (e.g., feeling irritable, feeling stressed) rather than a wider range of lapse predictors (e.g., feeling confident in one's ability to stay quit, feeling motivated to stay quit). This may have implications for JITAI effectiveness down the line and merits further investigation in experimental studies.

A 'warm start' approach is commonly used by adaptive intervention designers, as it enables learning algorithms to make decisions when encountering 'unknown states' [29]. Although the individual-level algorithms performed better than the group-level algorithm in the present study (albeit on a smaller number of participants), it should be noted that such an approach may not be theoretically feasible or desirable in the context of smoking cessation. For an individual-level approach to hit the ground running, a pre-intervention algorithm training period would first need to be implemented [27,57]. In the context of smoking cessation, however, where pre-quit predictors of *ad libitum* smoking can differ in magnitude and/or direction from post-quit lapse predictors [63,64], it may not be theoretically desirable for a JITAI to start from an individual approach (although see Naughton et al., 2023 for an example of a JITAI with a pre-quit period in which participants were asked to record their smoking cues). JITAI intervention developers are therefore also encouraged to consider 'warm start' approaches such as hybrid algorithms, which start from a group-level algorithm and subsequently move to an individual-level algorithm as more data from the post-quit period become available.

## Acceptability for participants

We found that individual-level and hybrid algorithms performed better than group-level algorithms, particularly when incorporating the sensor data. However, it remains unclear if many smokers would find it acceptable to share sensor data with researchers and practitioners (particularly for JITAIs where enrolment is fully remote), with such data being intrusive into people's lives. We are in the process of conducting focus groups, drawing on user centred design principles, to further explore the acceptability of using sensor data to underpin a future smoking cessation JITAI.

The large number of EMAs per day in the present study (i.e., 16 hourly EMAs/day) is highly unlikely to be feasible in a real-world setting. We plan to explore implications for algorithm performance when varying the number of EMAs (e.g., 8 or 4 versus 16 daily EMAs) in secondary analyses. The number of EMAs also has implications for the number of interventions that may be sent to each participant per day, including how many of these would likely be false positives based on the algorithm performance. This would need to be carefully considered going forwards.

## Scalability and technical feasibility for intervention developers

Worldwide, it has been estimated that there are currently more than 200 million smartwatches in use to monitor health-related information [65]. Although the inclusion of smartwatches (or other wearable devices) may be suitable for smaller scale JITAI implementations with 100s rather than 1000s of participants, it may not currently be feasible as part of larger-scale public health initiatives. It would not be economically viable to provide 1000s of participants with a free smartwatch and it would not be ethical to only offer the JITAI to participants with access to their own wearable device, as this would further perpetuate the digital divide. However, in the future, libraries and healthcare systems could potentially be set up as structures where smartwatches could be 'checked out' for a time-limited period, as one does not need to wear these in perpetuity to get the benefit of having it help with a smoking cessation attempt. These scalability and accessibility issues should be carefully considered by JITAI developers.

The data pre-processing required for the sensor data feature extraction and the matching of the features to the EMAs can be onerous. In addition, intervention developers need to factor in the server and mobile phone requirements relating to data storage, pre-processing, and the continuous re-estimation of the machine learning algorithms. For example, if it is desirable for computations to run on the server side rather than locally on participants' mobile phones, this will have implications for internet connectivity requirements and mobile phone battery drain. Some of the technical aspects of JITAI delivery can usefully be considered within the broader frame of building bespoke versus leveraging existing digital infrastructure(s). For example, the m-Path platform (which was used in the present study to deliver the EMAs) is expanding its functionalities to incorporate wearable sensor data and facilitate micro-randomisation [66]. In addition, mindLAMP (an open source platform developed by researchers at Harvard University) [67] and the Insight[TM] mHealth Platform (developed by researchers at the University of Oklahoma Health Sciences Center; https://healthpromotionresearch.org/MHealth#965585768-jitai-features) were built specifically to facilitate JITAI delivery. Depending on whether intervention developers are building a standalone JITAI component (versus implementing the JITAI within a wider, multi-component intervention), some of the above-mentioned technical issues can potentially be mitigated through leveraging existing digital infrastructure(s).

## Strengths and limitations

This study was strengthened by the prompted study design, sending frequent EMAs in people's daily lives, and being one of the first studies in the smoking cessation domain which used passively sensed data to distinguish lapse from non-lapse events. We also used a novel approach to feature extraction, iteratively testing different prediction distances and time windows [55]. Our interdisciplinary team of behavioural and data scientists leveraged Open Science principles to enable the re-use of the data, code, and study materials [68].

This study also had several limitations. First, as is typical for intensive longitudinal studies and particularly those involving passive sensing, missing data were common, which limits the conclusions that can be drawn [30,69]. For example, there may have been systematic differences between more and less adherent participants. However, a recent study found that wearable device adherence among insufficiently-active young adults appeared to be independent of identity and motivation for physical activity [70]. We tried to mitigate the impact of missing data through imputation with a univariable Kalman filter. However, the performance of different imputation techniques in relation to supervised machine learning algorithm training and testing is, to the best of our knowledge, underexplored. For example, multiple imputation techniques typically applied to intensive longitudinal data (e.g., multivariate imputation by chained equations) appear suboptimal when using $k$-fold cross-validation and parameter tuning. We plan to explore this in future work.

Second, we had planned to examine heart rate variability in this study. Due to the low sampling frequency of the commercial sensor used (i.e., a Fitbit Charge 4), however, we had to focus instead on heart rate. We found initial evidence that features relating to heart rate (e.g., change in slope) may be important predictors of lapses; however, further research is needed to replicate and extend these early results, drawing on techniques from the signal processing literature to filter out noise pertaining to, for example, physical activity or nicotine use [33]. Sensitivity analyses were conducted in which segments where participants indicated that they were walking or exercising, or consuming caffeine or nicotine were removed. However, further research is required to better understand what aspects of heart rate are useful for lapse prediction, under what conditions, and why–such work has been referred to as digital biomarker discovery [71].

Third, we had planned to vary the proportions of data used for training and testing of the hybrid algorithms (e.g., 40% training, 60% testing). However, we opted against running these additional sensitivity analyses due to the reflection that if moving ahead with a hybrid algorithm, it would be more efficient if the switch from group- to individual-level prediction was operated based on a performance threshold rather than precise data availability (as the latter is likely more sensitive to participant-specific fluctuations). However, the reliance on specific performance metrics–without considering a wider range of indicators such as the F1 score or precision-recall curve–constitutes another limitation of the present study which merits attention in future research. The selected performance metrics (i.e., sensitivity, specificity, the area under the receiver operator characteristic curve) cannot be computed when the test set contains only one of the two classes. This was the case for a non-trivial proportion of participants who reported 0% lapses and hence limited the number of participants for whom the algorithms could be constructed. Future research should consider using the Matthews Correlation Coefficient for imbalanced datasets [72], in addition to devising appropriate cut-offs for use in the context of a smoking lapse estimator, which are currently lacking. Related to this, research shows that supervised algorithm performance is impacted by several factors, including the sample size, the number of predictors, and other characteristics of the data, including its quality [73]. The sample size for the individual-level analyses in the present study (i.e., 160

observations per individual) was larger than in similar published studies [74,75] and in line with the larger (rather than smaller) sample sizes evaluated in simulation studies [73], which lends confidence to the small number of individual-level algorithms that could be constructed. As the abovementioned class imbalances (i.e., a non-trivial proportion of participants who reported 0% lapses) were likely a greater concern than the sample size in the present study, additional methodological research is needed to better understand how sample size and characteristics of the data influence both group- and individual-level algorithm performance.

Finally, the current results are limited to London-based individuals who were willing to participate in a research study with high measurement frequency. External algorithm validation in different populations (e.g., individuals with lower socioeconomic position) and settings (e.g., individuals residing in rural or smaller urban environments) is therefore needed. We encourage such explorations using our openly available R code in future research.

## Avenues for future research

First, to further reduce participant measurement burden, it may be useful to move from EMAs to micro-EMAs (i.e., very brief surveys delivered via smartwatches, which take a few seconds rather than minutes to complete) [76]. Another approach may be to leverage computerised adaptive testing [77] or so-called 'JITAI EMAs', which both aim to reduce the number of EMA items needed to classify an individual's current state [78].

Second, the predictive power of additional passive sensors could fruitfully be explored going forwards, such as the Global Positioning System receivers on most smartphones or text/ call logs [30,55]. However, such sensors further increase the level of intrusiveness. In addition, others have reported that data pre-processing of Global Positioning System data can be onerous [30].

Third, it would be useful to consider participant availability and receptivity alongside vulnerability in future JITAIs for smoking cessation. For example, in the JITAI 'HeartSteps', participants were considered unavailable if they were currently driving, did not have an active internet connection, they manually turned off the intervention, or were walking within 90 seconds of a decision point [19].

Fourth, although this study aimed to devise a method to help researchers determine when to intervene (i.e., identifying decision points and tailoring variables), it did not address the issue of how to intervene at moments of vulnerability, which is typically investigated through experimental methods. The average treatment effect with regards to near-term, proximal outcomes (e.g., the effect of delivering a tailored intervention at moments of vulnerability compared with not delivering an intervention on the risk of lapsing within the next few hours), can be studied through the micro-randomised trial design [79]. If the goal is to understand what works for a given individual, the system identification experimental design can beneficially be deployed [18]. Another approach still, which provides less insight into how to intervene at moments of vulnerability, would be to conduct a two-arm pilot RCT, estimating the average treatment effect with regards to a distal outcome (e.g., the effect of providing tailored interventions at moments of vulnerability compared with not delivering such tailored interventions on the odds of smoking cessation at a one-month follow-up assessment).

Taken together, the results from this study highlight several theoretical, methodological, and practical considerations for JITAI developers which, to the best of our knowledge, have not been previously articulated within a single paper. We set out to select a best-performing and feasible algorithm to predict lapses and therefore identify moments of predicted need for intervention by a subsequent JITAI. Although the predictive modelling approach evaluated here shows promise, rather than at this stage acting as a plug-and-play method for the

identification of decision points and tailoring variables in future JITAIs, researchers and developers are encouraged to carefully consider: the expected distribution of the outcome variable within individuals over time (and if information is lacking, it may be useful to first conduct a series of descriptive analyses to better understand the phenomenon prior to algorithm training and testing), appropriate algorithm performance metrics to focus on given the distribution of the outcome variable, the feasibility of passive sensing within the target population, ways of managing missing data, and the robustness of different algorithm interpretability methods. With further refinements, the predictive modelling approach deployed in the present study may become a useful off-the-shelf tool for JITAI developers.

## Conclusion

High-performing group-level lapse prediction algorithms had variable performance when applied to unseen individuals. Individual-level and hybrid algorithms had improved performance, particularly when incorporating sensor data for participants with sufficient wear time. Hybrid algorithms could be constructed for about twice as many individuals as the individual-level algorithms. Multiple success criteria (e.g., acceptability, scalability, technical feasibility) need to be balanced against algorithm performance in the JITAI development and implementation process.

## Supporting information

**S1 Table. Online screening survey.**
(DOCX)

**S2 Table. Additional baseline survey questions for eligible participants.**
(DOCX)

**S3 Table. EMA items.**
(DOCX)

**S4 Table. Model performance for the four algorithm types trained and tested on the nine different predictor distance-time window combinations.**
(DOCX)

**S1 Fig. Area under the receiver operating characteristic curve (AUC) estimates and accompanying 95% confidence intervals for the best-performing group-level models (without sensor data).**
(DOCX)

**S2 Fig. Variable importance plots for the best-performing group-level algorithms. The variable importance score does not indicate the direction of the relationship between the predictor and outcome variable.**
(DOCX)

**S3 Fig. Proportion of participants with each of the predictor variables in their top 10 ($n$ = 15). For clarity, predictor variables that were not included in a single participant's top 10 are not displayed.**
(DOCX)

**S4 Fig. Area under the receiver operating characteristic curve (AUC) estimates and accompanying 95% confidence intervals for the best-performing algorithms (with sensor data).**
(DOCX)

**S5 Fig. Variable importance plots for the best-performing group-level algorithms (with sensor data).** The variable importance score does not indicate the direction of the relationship between the predictor and outcome variable.
(DOCX)

**S6 Fig. Proportion of participants with each of the predictor variables in their top 10 ($n$ = 11). For clarity, predictor variables that were not included in a single participant's top 10 are not displayed.**
(DOCX)

**S7 Fig. Plot of the area under the receiver operating characteristic curve (AUC) estimate for each of the group-level algorithms (sensitivity analysis).**
(DOCX)

**S8 Fig. Variable importance plots for the best-performing group-level algorithms (sensitivity analysis). The variable importance score does not indicate the direction of the relationship between the predictor and outcome variable.**
(DOCX)

**S9 Fig. Variable importance plot for the best-performing group-level Random Forest algorithm (sensitivity analysis).** The variable importance score does not indicate the direction of the relationship between the predictor and outcome variable.
(DOCX)

**S10 Fig. Frequency distributions of the performance metrics of interest (i.e., accuracy, sensitivity, specificity, AUC) for the best-performing individual-level algorithms (n = 31; sensitivity analysis).** The shaded grey areas represent the prespecified thresholds for acceptable accuracy (0.70), sensitivity (0.70), specificity (0.50), and AUC (0.50). The solid vertical lines represent the median.
(DOCX)

**S11 Fig. Proportion of participants with each of the predictor variables in their top 10 ($n$ = 31; sensitivity analysis). For clarity, predictor variables that were not included in a single participant's top 10 are not displayed.**
(DOCX)

**S12 Fig. Area under the receiver operating characteristic curve (AUC) estimates and accompanying 95% confidence intervals for the best-performing group-level models with sensor data (without and with the removal of timepoints with potential confounders).**
(DOCX)

**S13 Fig. Proportion of participants with each of the predictor variables in their top 10 ($n$ = 9). For clarity, predictor variables that were not included in a single participant's top 10 are not displayed.**
(DOCX)

## Author Contributions

**Conceptualization:** Olga Perski, Stephanie P. Goldstein, Eric Hekler, Jamie Brown.

**Data curation:** Olga Perski, David Simons.

**Formal analysis:** Olga Perski, David Simons.

**Funding acquisition:** Jamie Brown.

**Investigation:** Olga Perski, Dimitra Kale, Corinna Leppin, Tosan Okpako.

**Methodology:** Olga Perski, Stephanie P. Goldstein.

**Project administration:** Olga Perski, Dimitra Kale.

**Visualization:** David Simons.

**Writing – original draft:** Olga Perski.

**Writing – review & editing:** Dimitra Kale, Corinna Leppin, Tosan Okpako, David Simons, Stephanie P. Goldstein, Eric Hekler, Jamie Brown.

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
