## [Decision Letter · Decision Letter 0]

13 Feb 2024

PDIG-D-24-00003

Supervised Machine Learning to Predict Smoking Lapses from Ecological Momentary Assessments and Sensor Data: Implications for Just-in-Time Adaptive Intervention Development

PLOS Digital Health

Dear Dr. Perski,

Thank you for submitting your manuscript to PLOS Digital Health. After careful consideration, we feel that it has merit but does not fully meet PLOS Digital Health's publication criteria as it currently stands. Therefore, we invite you to submit a revised version of the manuscript that addresses the points raised during the review process.

Please submit your revised manuscript within 60 days Apr 13 2024 11:59PM. If you will need more time than this to complete your revisions, please reply to this message or contact the journal office at digitalhealth@plos.org. Please include the following items when submitting your revised manuscript:

We look forward to receiving your revised manuscript.

Kind regards,

Calvin Or, PhD

Section Editor

PLOS Digital Health

Journal Requirements:

Additional Editor Comments (if provided):

Reviewers' comments:

Reviewer's Responses to Questions

**Comments to the Author**

1. Does this manuscript meet PLOS Digital Health’s publication criteria? Is the manuscript technically sound, and do the data support the conclusions? The manuscript must describe methodologically and ethically rigorous research with conclusions that are appropriately drawn based on the data presented.

Reviewer #1: Yes

Reviewer #2: Yes

Reviewer #3: Partly

2. Has the statistical analysis been performed appropriately and rigorously?

Reviewer #1: Yes

Reviewer #2: Yes

Reviewer #3: Yes

3. Have the authors made all data underlying the findings in their manuscript fully available (please refer to the Data Availability Statement at the start of the manuscript PDF file)?

Reviewer #1: Yes

Reviewer #2: Yes

Reviewer #3: Yes

4. Is the manuscript presented in an intelligible fashion and written in standard English?

Reviewer #1: Yes

Reviewer #2: Yes

Reviewer #3: Yes

5. Review Comments to the Author

Reviewer #1: This paper investigates the feasibilities of using supervised machine learning algorithm that uses ecologically momentary assessment and wearable sensor data to identify the moments of lapses among the adult smokers attempting to quit. The results show that the individual-level and hybrid algorithms had improved the performance, particularly when incorporating sensor data for participants with sufficient wear time. However, high-performing group-level lapse prediction algorithms without and with sensor data had variable performance when applied to out-of-sample individuals.

Overall, this is a well-written paper. The workflow and the findings of the study are clear. It is easy to follow the manuscript. I would list out some key concerns as below.

# Research Contributions

I felt the research contribution of this work seems to be slightly vague. I would also suggest the authors to include a brief contribution claim at the end of the introductory section such as the key findings, instead of listing out the objective of the study? The current manuscript, I personally feel, looks more descriptive (like a technical/UX report), instead of a paper. While I could see the insightful findings from the discussion, it would be possibly helpful to claim the key contribution at the very beginning?

# Study Duration and the Participants

I am wondering if the 10 days are sufficient to verify the claimed finding, which seems somewhat short. I am also wondering if there is any reporting related to how long the recruited participants have of the smoking history?

# Chosen of the Algorithms

I am wondering if the authors have tried out other machine learning or deep learning algorithms? I felt a simple vanilla ANN with few number of hidden layers seems to be a nice candidate model to evaluate? I don’t think the computational overhead would be too heavy to be deployed on the edge devices.

# Structures, Formatting and Misc.

Page 4 – 6: it looks the introduction seems to be mixed with the related work/background section. I personally would suggest the author to see if there are any possibilities to separate them, to enhance the readability. The key focus is to justify the fundamental different to the existing related works, such as (11, 19, 23 - 26)

In the penultimate paragraph of the page 5, the format of the in-line citation seems not to be consistent. I would suggest the authors use the same numerical citation format. 

Figure 1 seems to be “over-cropped” at the bottom? I would also suggest to revise the caption. It is quite vague by describing the figure as “participant flow chart”?

Font size of the Figure 2, 4, 5, and 7 possibly needs to be increased.

Reviewer #2: This is a comprehensive paper that does a great job of highlighting the background/significance of this study and overviewing its findings. There are some elements that can be improved to make it an excellent candidate for publication.

Decision: Accept pending revisions

Main comments:

1. The study includes an interesting combination of data sources (passive data from smartwatches and self-reported data from surveys)

2. The introduction provides a good explanation for why JITAIs are particularly relevant for smoking cessation. The authors include a great discussion of limitations of previously existing JITAIs with inclusion of specific examples

3. An overview table summarizing key limitations of prior JITAIs compared to this one could supplement the introduction beyond the text that is included, making it clearer how this study advances on others

4. Methods include a comprehensive description of the study design. However, I do not see a description of the overall timeframe for the study – this information should be included. For instance, I wonder how seasonality / time of year might affect your results if participant data were collected in different months.

5. The inclusion of power analysis to evaluate needed sample sizes for statistical analysis is great and ameliorates my concerns about sample size for group-level analysis. However, I am concerned about sample size limitations for the individual-level analysis. The histograms in Figures 4 and 7 don’t seem very informative because the sample size is so low. Given the results, I would suggest spending include more figures/discussion for the group-level analysis and less on the individual-level analysis.

6. AUC results (Figures 2 and 5) for group-level analysis could also use a box plot or histogram (similar to Figures 4 and 7) that includes standard deviations to show how close the AUCs from the different models are to one another

7. The authors include a good discussion of using cravings vs. lapses as an outcome and warm start approaches

8. Introduction could use a few more sentences clarifying how this study is related to identifying best intervention time over best intervention method. This point is included in the Discussion, but it should be made clear earlier on.

9. Discussion could use an interpretation of the ML results – why do specific models work better than others for the various data settings?

10. Discussion could also use an interpretation of the variable importance results – why might certain variables matter more than others?

11. Discussion could use more of a description of limitations / confounders and reproducibility in the future

Population applicability – It should be mentioned that current results are limited to individuals from a London-based population and need to be explored in different populations with different environments and different socioeconomic status

12. Discussion could use a description of how to translate/productionalize this work – I.e., a few more sentences about how these results might inform future JITAI development.

Minor comments:

1. GitHub is organized well and provides a good overview of the code for analysis. Commenting in scripts could be improved a bit further, and the README file could include a better overview of the importance of each script.

2. Small grammar error: “Finally, although this study aimed to devise a method to help researchers determine when to intervene, it not address the issue of how to intervene at moments of vulnerability”

3. Introduction: “Other approaches to the identification of JITAI decision points and tailoring variables (not expanded on further in this paper)…” This is a long sentence and is hard to follow. Break it down into smaller pieces.

4. The font size for Figures 2 to 7 could be increased to improve readability

Reviewer #3: It is a well written manuscript, providing a good level of technical information and details with the methodology that is acceptable. The reason behind using simple machine learning models is reasonable and the results show a good (and acceptable in some settings) performance. Although evaluating area under ROC curve can be a good metric for model and performance evaluation, it would be preferred to add more metrics which are useful and suitable for clinical analysis applications such as precision-recall and F1-score. It would improve the credibility of the results and help with the comparison and evaluation of AUC for the model performance. 

Also summarising the introduction to some extent might help with the flow and story of the work.

6. PLOS authors have the option to publish the peer review history of their article (what does this mean?). If published, this will include your full peer review and any attached files.

**Do you want your identity to be public for this peer review?** For information about this choice, including consent withdrawal, please see our Privacy Policy.

Reviewer #1: No

Reviewer #2: No

Reviewer #3: No

---

## [Decision Letter · Decision Letter 1]

9 Jun 2024

PDIG-D-24-00003R1

Supervised Machine Learning to Predict Smoking Lapses from Ecological Momentary Assessments and Sensor Data: Implications for Just-in-Time Adaptive Intervention Development

PLOS Digital Health

Dear Dr. Perski,

Thank you for submitting your manuscript to PLOS Digital Health. After careful consideration, we feel that it has merit but does not fully meet PLOS Digital Health's publication criteria as it currently stands. Therefore, we invite you to submit a revised version of the manuscript that addresses the points raised during the review process.

Please submit your revised manuscript within 60 days Aug 08 2024 11:59PM. If you will need more time than this to complete your revisions, please reply to this message or contact the journal office at digitalhealth@plos.org. Please include the following items when submitting your revised manuscript:

We look forward to receiving your revised manuscript.

Kind regards,

Calvin Kalun Or, PhD

Section Editor

PLOS Digital Health

Journal Requirements:

1. Please provide separate figure files in .tif or .eps format only and remove any figures embedded in your manuscript file. Please also ensure that all files are under our size limit of 10MB.

2. We have noticed that you have uploaded Supporting Information files, but you have not included a list of legends. Please add a full list of legends for your Supporting Information files after the references list.

Additional Editor Comments (if provided):

A reviewer provided a useful comment in the last review round: “… if the authors have tried out other machine learning or deep learning algorithms? I felt a simple vanilla ANN with few number of hidden layers seems to be a nice candidate model to evaluate? …” The comment is valid, and I suggest the authors to apply the relevant algorithms to see whether better prediction performance would be obtained, rather than pointing out the issue as a limitation.

It is also important to note the comment about “how seasonality / time of year might affect the results”. Again, I suggest authors to consider conducting an analysis to address the comment rather than citing it as a limitation.

Reviewers' comments:

Reviewer's Responses to Questions

**Comments to the Author**

1. If the authors have adequately addressed your comments raised in a previous round of review and you feel that this manuscript is now acceptable for publication, you may indicate that here to bypass the “Comments to the Author” section, enter your conflict of interest statement in the “Confidential to Editor” section, and submit your "Accept" recommendation.

Reviewer #4: (No Response)

2. Does this manuscript meet PLOS Digital Health’s publication criteria? Is the manuscript technically sound, and do the data support the conclusions? The manuscript must describe methodologically and ethically rigorous research with conclusions that are appropriately drawn based on the data presented.

Reviewer #4: Yes

3. Has the statistical analysis been performed appropriately and rigorously?

Reviewer #4: Yes

4. Have the authors made all data underlying the findings in their manuscript fully available (please refer to the Data Availability Statement at the start of the manuscript PDF file)?

Reviewer #4: Yes

5. Is the manuscript presented in an intelligible fashion and written in standard English?

Reviewer #4: Yes

6. Review Comments to the Author

Reviewer #4: This study (n=38 smokers attempting to quit; 7% reported a lapse) examined the use of EMA and Fitbit data collected over 10-days to predict smoking lapse (and in sensitivity analyses: craving dichotomously coded). Individual and hybrid (combination of individual and group-level algorithms) models performed better than group-level models, and sensor data added predictive info above EMA data alone. However, the addition of sensor data needs to be considered against feasibility because fewer individuals provided both types of data. 

The revision is responsive to comments: Introduction was re-organized and includes a summary table of prior relevant research; Discussion addresses limitations regarding the models tested (e.g., no deep learning models), limited model performance metrics examined, and limitations to generalizability of results; and the need for caution in interpreting results when sample size is low and outcome/data are sparse.

Some minor additional comments could be considered:

1. Methods section mentions potential confounding factors affecting heart rate, which were assessed in EMA (e.g., caffeine use, physical activity). However, the way in which these possible confounds were taken into account in the analyses when examining heart rate as a predictor of lapse is unclear, and could be addressed (e.g., in supplementary materials) since the confounds might contribute an important source of error.

2. Page 10: “participants with insufficient sensor wear time were removed from the analytical sample, defined as >20% adherence…” The phrasing of this statement seems to suggest that the definition refers to “insufficient” wear time. It might be clearer to rephrase this to, e.g.: “participants with sufficient sensor wear time were defined as having >20% adherence…”

7. PLOS authors have the option to publish the peer review history of their article (what does this mean?). If published, this will include your full peer review and any attached files.

**Do you want your identity to be public for this peer review?** For information about this choice, including consent withdrawal, please see our Privacy Policy. 

Reviewer #4: No

---

## [Decision Letter · Decision Letter 2]

17 Jul 2024

PDIG-D-24-00003R2

Supervised Machine Learning to Predict Smoking Lapses from Ecological Momentary Assessments and Sensor Data: Implications for Just-in-Time Adaptive Intervention Development

PLOS Digital Health

Dear Dr. Perski,

Thank you for submitting your manuscript to PLOS Digital Health. After careful consideration, we feel that it has merit but does not fully meet PLOS Digital Health's publication criteria as it currently stands. Therefore, we invite you to submit a revised version of the manuscript that addresses the points raised during the review process.

Please submit your revised manuscript within 60 days Sep 15 2024 11:59PM. If you will need more time than this to complete your revisions, please reply to this message or contact the journal office at digitalhealth@plos.org. Please include the following items when submitting your revised manuscript:

We look forward to receiving your revised manuscript.

Kind regards,

Calvin Or, PhD

Section Editor

PLOS Digital Health

Journal Requirements:

Additional Editor Comments (if provided):

Reviewers' comments:

Reviewer's Responses to Questions

**Comments to the Author**

1. If the authors have adequately addressed your comments raised in a previous round of review and you feel that this manuscript is now acceptable for publication, you may indicate that here to bypass the “Comments to the Author” section, enter your conflict of interest statement in the “Confidential to Editor” section, and submit your "Accept" recommendation.

Reviewer #5: All comments have been addressed

2. Does this manuscript meet PLOS Digital Health’s publication criteria? Is the manuscript technically sound, and do the data support the conclusions? The manuscript must describe methodologically and ethically rigorous research with conclusions that are appropriately drawn based on the data presented.

Reviewer #5: Yes

3. Has the statistical analysis been performed appropriately and rigorously?

Reviewer #5: No

4. Have the authors made all data underlying the findings in their manuscript fully available (please refer to the Data Availability Statement at the start of the manuscript PDF file)?

Reviewer #5: Yes

5. Is the manuscript presented in an intelligible fashion and written in standard English?

Reviewer #5: No

6. Review Comments to the Author

Reviewer #5: The authors have responded well to the previous reviewers' comments and revised the article. However, there are still areas in the text that need to be revised.

1. Grammatical problems. The last sentence of the first paragraph of "Data analysis," "(i.e., cravings at t0 was used to predict lapse incidence at t1)." is inconsistent. Please revise the sentence and confirm the whole text for no grammatical or spelling problems.

2. The first sentence in the "Model training and testing" of the "Data analysis" section： "Model training and testing was performed through k-fold cross-validation (55), with k set to 10. For each iteration (or fold), models were trained on 80% of the data and tested on the remaining 20% of the data." The expression of the k-fold cross-validation used is unclear and ambiguous. The current statement does not meet the definition of k-fold cross-validation. If a stratified k-fold method was used, please indicate the basis or reasons for such stratification.

3. Table 2 shows that the “Welch Two Sample t-test” and “Pearson's Chi-squared test” were employed. Please supplement the statistical methods used in the data analysis and the purpose.

7. PLOS authors have the option to publish the peer review history of their article (what does this mean?). If published, this will include your full peer review and any attached files.

**Do you want your identity to be public for this peer review?** For information about this choice, including consent withdrawal, please see our Privacy Policy. 

Reviewer #5: No

---

## [Editor Report · Decision Letter 3]

28 Jul 2024

Supervised Machine Learning to Predict Smoking Lapses from Ecological Momentary Assessments and Sensor Data: Implications for Just-in-Time Adaptive Intervention Development

PDIG-D-24-00003R3

Dear Dr Perski,

We are pleased to inform you that your manuscript 'Supervised Machine Learning to Predict Smoking Lapses from Ecological Momentary Assessments and Sensor Data: Implications for Just-in-Time Adaptive Intervention Development' has been provisionally accepted for publication in PLOS Digital Health.

Best regards,

Calvin Or, PhD

Section Editor

PLOS Digital Health